# Association between Endocrine Therapy and Weight Gain after Breast Cancer Diagnosis among Japanese Patients: A Retrospective Cohort Study

**DOI:** 10.3390/medsci9030050

**Published:** 2021-07-12

**Authors:** Koki Okumatsu, Hideko Yamauchi, Rina Kotake, Masahiko Gosho, Yoshio Nakata

**Affiliations:** 1Graduate School of Comprehensive Human Sciences, University of Tsukuba, 1-1-1 Tennodai, Tsukuba 305-8577, Japan; s1830360@s.tsukuba.ac.jp; 2Department of Breast Surgical Oncology, St. Luke’s International Hospital, 9-1 Akashicho, Chuo-ku, Tokyo 104-8560, Japan; hideyama@luke.ac.jp (H.Y.); rinakotakemd@gmail.com (R.K.); 3Department of Biostatistics, Faculty of Medicine, University of Tsukuba, 1-1-1 Tennodai, Tsukuba 305-8575, Japan; mgosho@md.tsukuba.ac.jp; 4Faculty of Health and Sport Sciences, University of Tsukuba, 1-1-1 Tennodai, Tsukuba 305-8577, Japan

**Keywords:** cancer survivors, body weight change, exercise, eating

## Abstract

Background: This study aimed to investigate the association between endocrine therapy and weight gain with a history of breast cancer. Methods: This was a retrospective cohort study. Recruited patients consisted of those receiving endocrine therapy and those not receiving endocrine therapy. Weight at diagnosis was collected from medical records, and current body mass was measured using a digital scale (time since diagnosis was 4.0 ± 1.9 years). Moreover, we measured current physical activity using an accelerometer and dietary intake using a validated questionnaire. The primary analysis was a *t*-test for the body-mass change after diagnosis between the two groups. Results: We recruited 300 patients and collected data from 292. Mean weight gain after diagnosis was 1.3 ± 3.9 kg, and the change in body mass of patients taking endocrine therapy (1.3 ± 4.0 kg) was not significantly different from that of patients not taking endocrine therapy (1.4 ± 3.8 kg, *p* = 0.92). There was no association of endocrine therapy, physical activity, and dietary intake with a 5% weight gain after adjusting confounding factors (e.g., breast cancer stage and chemotherapy). Conclusions: Caution is required with generalization because of sampling bias and ethnic differences.

## 1. Introduction

Breast cancer is the most common cancer and the leading cause of cancer-related death both in Asia and worldwide [1,2]. A previous study reported that breast cancer incidence is increasing in Asia [3]. A common physical issue among breast cancer patients is weight gain during and after treatment [4]. Irwin et al. [5] reported that 68% of patients with breast cancer gained weight within 3 years of their diagnosis, and a literature review showed that the mean weight gain ranged from 2.5 kg to 6.2 kg during the first year after diagnosis [6]. Excessive weight gain and the subsequent onset of obesity contribute to an increased risk of all-cause mortality and breast cancer recurrence [7]. Therefore, weight management plays an essential role in avoiding poor health outcomes among breast cancer patients [4]. 

Endocrine therapy may be associated with weight gain. Gibb et al. reported that endocrine therapy is associated with significantly lower insulin sensitivity and greater percentage of body fat compared with an age-matched control group [8]. The detailed mechanism of this association is unclear, but Gibb et al. suggested that endocrine therapy is related to estrogen deficiency. The change in hormonal balance may be related to weight gain. According to Nestoriuc et al. [9], more than 50% of patients with breast cancer reported that weight gain was one of the side effects of endocrine therapy. Nyrop et al. [10] stated that the type of endocrine therapy was not related to weight gain. The relationship between endocrine therapy and weight gain remains incompletely understood. However, other factors such as physical activity and dietary intake might also be related to weight gain. Brown et al. [11] reported that 30% of breast cancer patients reduced their physical activity after the initiation of treatment, and low physical activity could be related to weight gain [5]. Chen et al. [12] reported that patients who had a higher consumption of total dietary intake, including meat, fish, cruciferous vegetables, and soy food, gained more body mass than those with a lower intake. Since the association between these factors and weight gain is complex, further studies are required to simultaneously assess their effect in patients undergoing endocrine therapy.

Although some studies have focused on the association between endocrine therapy, physical activity, and dietary intake with weight gain, they had certain limitations in measuring physical activity and dietary intake. Most previous studies used questionnaires to evaluate physical activity [13,14,15]; however, these methods likely overestimated the amount of physical activity compared with an accelerometer [16,17]. Most previous dietary surveys were reported from Europe and the U.S.; however, the questionnaires on dietary history differ among Europe, the U.S., and Asia [18,19,20,21]. No study has used both accelerometers and specific nutritional questionnaires at the same time to investigate the factors associated with weight gain among breast cancer patients in an Asian population.

Therefore, the present study aimed to test the hypothesis that the change in body mass following breast cancer diagnosis among patients receiving endocrine therapy was different from those not receiving endocrine therapy. Moreover, we examined the association between endocrine therapy, objectively measured physical activity, and dietary intake with weight gain among Japanese breast cancer patients. Findings from this study might contribute towards the development of a rationale for implementing a weight-management program for breast cancer survivors.

## 2. Patients and Methods

### 2.1. Study Design and Setting

The present study was a retrospective cohort study to examine the association between endocrine therapy, physical activity, and dietary intake and the change in body mass after breast cancer diagnosis among Japanese patients. All patients gave their informed consent for inclusion before they participated in this study. This study was conducted in accordance with the Declaration of Helsinki, and the protocol was approved by the Ethics Committee of the St. Luke’s International Hospital (18-R059 approved on 18 October 2018) and the University of Tsukuba (30-22R approved on 29 October 2018). A grant from the Japanese Society of Test and Measurement in Health and Physical Education partly supported this study.

### 2.2. Eligibility Criteria

We recruited patients who were diagnosed with breast cancer from December 2018 to October 2019 at St. Luke’s International Hospital (Tokyo, Japan). The inclusion criteria of this study were: (1) women aged between 20 and 64 years; (2) no severe cognitive impairment (e.g., dementia); and (3) those not diagnosed with metastatic cancer, excluding those with lymphatic metastases. We used medical records to determine the presence of cognitive impairment. The recruited patients consisted of those taking endocrine therapy and those not taking endocrine therapy. The former had received endocrine therapy for at least 2 years, and the latter had been diagnosed with breast cancer at least 2 years previously and had not received endocrine therapy. This study aimed to investigate the association between endocrine therapy and weight gain, so we investigated these relationships irrespective of the use of chemotherapy and radiation therapy. Endocrine therapy consisted of tamoxifen, aromatase-inhibitor, and a combination of these.

### 2.3. Study Process

Potential patients received a brief explanation of this study from a physician specializing in breast surgical oncology and received a detailed description from the study staff at the hospital. If patients decided to join this study, they signed an informed consent form. Patients were included in either the group taking endocrine therapy or the group not taking endocrine therapy to examine the difference in body mass between the two groups. Figure 1 shows the flowchart of this study.

### 2.4. Measurements

#### 2.4.1. Primary Outcome

The primary outcome was the change in body mass after diagnosis. With the patients in their underwear without shoes, we measured the current body mass to the nearest 0.1 kg using a digital scale (TB-150; Tanita, Tokyo, Japan). The patients were not instructed to refrain from consuming any food or drinks before having their body mass measured. This was because the priority reason for visiting the hospital by the patients was to undergo their usual medical check-up at the hospital, and this study could not interfere with the usual medical check-up. However, the patients were instructed to have voided their bladder and bowels before having their body mass measured. We collected data on the body mass at diagnosis from the medical records of St. Luke’s International Hospital.

#### 2.4.2. Anthropometric and Medical History Measurements

We collected data on height, status of endocrine therapy, and menopausal status at diagnosis from the medical records of St. Luke’s International Hospital. We calculated body mass index (BMI) as weight (kg) divided by height (m) squared.

#### 2.4.3. Sociodemographic and Lifestyle Variables

The patients reported sociodemographic and lifestyle characteristics via self-administered questionnaires, including educational status, economic situation (household income), current smoking status, and current menopausal status (Appendix A, Appendix A). In addition, we collected data on the time since cancer diagnosis through the medical records.

#### 2.4.4. Physical Activity

We measured the current physical activity level using a validated triaxial accelerometer and the IPAQ questionnaire [22,23]. A triaxial accelerometer (Active style Pro HJA-750C; Omron Healthcare, Kyoto, Japan) can count steps and compute the intensity of physical activity from a published algorithm [22,24]. We instructed the patients to wear the accelerometer on their waists to estimate the intensity of physical activity based on metabolic equivalents (METs) during waking hours, except during water activities or participating in specific exercises for safety reasons (e.g., contact sports). Regarding cut points of physical activity intensity, the device was classified into three intensity categories based on METs: sedentary behavior (≤ 1.5 METs), low-intensity physical activity (1.6–2.9 METs), and moderate- to vigorous-intensity physical activity (MVPA; ≥ 3.0 METs). We collected the data in a 60-s period. If there were no acceleration signals for ≥ 60 consecutive minutes, we defined the period as “non-wear.” The daily record was regarded as valid when patients wore the device for at least 10 h/day. When valid days were less than 3 days, we excluded these records from the analysis. Finally, we calculated the mean for total daily minutes of sedentary behavior, low-intensity physical activity, and MVPA.

#### 2.4.5. Dietary Intake

We used the brief-type self-administered diet history questionnaire (BDHQ) to evaluate dietary intake, which is a short version of the self-administered diet history questionnaire (DHQ) that was previously validated in Japan [25]. The BDHQ is a 4-page fixed-portion questionnaire that assesses dietary intake by the reported consumption frequency of 58 different food and beverage items. It takes approximately 10 min to answer. Most of the food and beverage items are selected from the food list of the DHQ, as these items are popular in Japan. Standard portion sizes for women are based on the National Nutrition Survey of Japan and various recipe books for Japanese dishes [26]. The BDHQ includes not only the frequency of consumption of the selected foods but also the usual cooking methods and general dietary behavior.

#### 2.4.6. Sample Size

We used the G*power (Heinrich Heine University, North Rhine-Westphalia, Germany) [27] to calculate the required sample size. Based on a previous study by Heideman et al. [28], which examined the change in body mass among Dutch breast cancer patients for 5 years after diagnosis, those taking endocrine therapy gained 4.7 ± 6.3 kg, while those not taking endocrine therapy gained 1.7 ± 5.1 kg. Therefore, the calculated effect size was 0.5, whereas we set the effect size to 0.4 to estimate conservatively. We set the alpha error probability as 0.05 and the power as 0.8, and the required sample size was determined to be 200 in total (100 in each group). We set the target sample size at 250 (125 in each group), considering potential missing data.

After the study commencement, we noticed a difficulty in recruiting the two groups equally because the number of patients who did not receive endocrine therapy was smaller than those who received endocrine therapy. A previous study reported that the prevalence ratio of those taking endocrine therapy and those not receiving endocrine therapy might be 7:3 [29]. To maintain statistical power, we upwardly revised the target sample size and extended the recruitment to ensure that the number of patients not taking endocrine therapy reached 125.

#### 2.4.7. Statistical Methods

All analyses were conducted based on an intention-to-treat principle. We described all continuous variables at baseline as mean ± standard deviation (SD). To compare the difference between the two groups, we used an unpaired *t*-test and a chi-squared test for continuous and categorical variables, respectively. A test of normality was performed before the *t*-test. For items that did not conform to the normality test, we performed Mann–Whitney *U*-test and reported the median and quartiles. The primary outcome, the change in body mass after diagnosis, was analyzed using analysis of covariance (ANCOVA), including the treatment group (taking endocrine therapy or not) and breast cancer stage at diagnosis as factors, and the baseline body mass at diagnosis as a covariate.

Moreover, we conducted a logistic regression analysis to examine associations between taking endocrine therapy, physical activity, and dietary intake and gaining body mass after adjusting for confounding factors. Regarding logistic regression analysis, we categorized physical activity and dietary intake into quartiles based on the amount of MVPA and the dietary intake. We also calculated the percentage of weight change from the diagnosis of breast cancer as (current body mass − weight at diagnosis) / (weight at diagnosis) multiplied by 100. A gain in body mass was defined as a weight gain of more than 5% because of its clinical significance [30,31]. All tests were two-sided, and *p*-values < 0.05 were considered statistically significant. All statistical analyses were performed using SPSS version 26.0 (SPSS Inc., Chicago, IL, U.S.).

## 3. Results

We explained the study detail to 538 eligible patients, and 238 refused to participate because they were too busy with their work, family, and for other reasons. We did not officially count the reasons for the refusals, but most of these were related to being busy. We recruited 300 patients in total. Of these, eight were excluded due to missing data on the body mass at diagnosis. The remaining 292 patients included 172 who received endocrine therapy and 120 who did not receive endocrine therapy (Figure 1).

Table 1 shows the patient characteristics at the time of breast cancer diagnosis. All the patients underwent breast surgery. The average age and BMI at diagnosis were 46.6 ± 6.9 years and 21.3 ± 2.9 kg/m^2^, respectively, and there was no significant difference between the two groups. Although the body mass at diagnosis was higher in the patients taking endocrine therapy (mean difference, 1.7 kg; 95% confidence interval, 0.1–3.5 kg), BMI was not significantly different. Moreover, there was a significant difference in the breast cancer stage between the two groups (*p* < 0.01). Receipt of treatments (chemotherapy and radiation therapy) was not significantly different between the two groups.

Table 2 shows the patient characteristics at recruitment between the two groups. Overall, 13.4% of patients were underweight (BMI < 18.5), 70.2% were normal (18.5 to < 25), and 16.4% were overweight (≥ 25). The mean weight gain after diagnosis was 1.3 ± 3.9 kg, and 90 (30.8%) patients increased their body mass by more than 5%. There was no significant difference between the two groups in body mass change after diagnosis (endocrine therapy: 1.3 ± 4.0 kg; not taking endocrine therapy: 1.4 ± 3.8 kg; *t*-test: *p* = 0.92; ANCOVA: *p* = 0.34). BMI, objectively measured MVPA, and dietary intake also showed no significant difference between the two groups. However, the postmenopausal status was significantly different (*p* < 0.01).

Table 3 shows the odds ratio (OR) for 5% weight gain after diagnosis using valid data for 271 patients (159 received endocrine therapy and 112 did not receive endocrine therapy). After adjusting for confounding factors, there was no association between taking endocrine therapy, physical activity, or dietary intake and 5% weight gain. Age at diagnosis was significantly associated with 5% weight gain negatively in both crude (OR: 0.95; 95% confidence interval (CI): 0.92–0.989) and adjusted analysis (OR: 0.94; 95% CI: 0.88–0.99).

## 4. Discussion

The present study primarily examined changes in body mass after breast cancer diagnosis. We hypothesized that patients receiving endocrine therapy gained more body mass than those not receiving endocrine therapy; however, the results of the current study did not support our hypothesis. Moreover, a further logistic regression analysis showed no significant association between endocrine therapy, physical activity, or dietary intake and weight gain.

The previous associations between endocrine therapy and weight gain have been controversial. Nestoriuc et al. [9] reported that more than 50% of breast cancer patients regarded weight gain as one of the side effects of endocrine therapy. Similarly, Mortimer et al. [32] reported that 32% of breast cancer patients taking endocrine therapy reported weight gain as a severe symptom. The precise mechanism of weight gain among breast cancer patients receiving endocrine therapy is unclear, but the deficiency of estrogen might be related to weight gain. Endocrine therapy decreases estrogen levels, and Gibb et al. reported that women who received endocrine therapy for breast cancer had greater insulin resistance and higher body fat accumulation than the control group [8]. Therefore, these results indicate that lower estrogen level is associated with insulin resistance and weight gain. In contrast, Nyrop et al. [10] conducted a retrospective chart review and stated that the type of endocrine therapy was not related to weight gain. Kim et al. [33] also reported that taking endocrine therapy was not significantly associated with weight gain (OR: 0.69; 95% CI: 0.22–2.22). The present study results showed that there was no significant difference in weight gain between those who took endocrine therapy and those who did not. A racial difference might be one of the reasons why taking endocrine therapy was not related to weight gain. When we calculated the sample size, we referred to the Dutch study by Heideman et al. [28]. In their study, women who were taking endocrine therapy and chemotherapy had gained 4.7 ± 6.3 kg 5 years after diagnosis, but women who were not taking endocrine therapy (chemotherapy only) gained 1.7 ± 5.1 kg. In addition, a literature review that was mainly based on Western studies showed that the mean weight gain ranged from 2.5 kg to 6.2 kg during the first year after diagnosis [6]. However, the patients in the present study had only a slight increase in body mass of 1.3 ± 3.9 kg (taking endocrine therapy: 1.3 ± 4.0 kg; not taking endocrine therapy: 1.4 ± 3.8 kg; *p* = 0.92) 4 years after diagnosis. Gu et al. [34] evaluated weight change from diagnosis to 3 years post-diagnosis among Chinese breast cancer patients, 67.1% of whom received endocrine therapy. They reported that breast cancer patients had only a slight increase in body mass of 1.0 kg 3 years after diagnosis, and there was no significant relationship between endocrine therapy and weight change (*p* = 0.83). Compared with Western studies, Asian breast cancer patients might gain less weight, and there might be a racial difference in weight gain between Asian and Western breast cancer patients. Further studies are needed to address this issue.

Physical activity and dietary intake might also be associated with weight gain among breast cancer patients. Several previous studies in Asia have concurrently examined physical activity and dietary intake. We simultaneously assessed physical activity and dietary intake using objective and validated methods to examine the association between these factors and weight gain using a logistic regression model. The results showed no significant associations between physical activity or dietary intake and weight gain. However, we assessed physical activity and dietary intake at recruitment only, and we could not evaluate the change in these factors after diagnosis. Lynch et al. [35] conducted a cross-sectional study to assess the association between physical activity and waist circumference among U.S. breast cancer survivors. They measured physical activity using an accelerometer and reported that MVPA was negatively related to waist circumference (*β* = −9.805, 95% CI: −15.8 to −3.7). Kim et al. [33] conducted a retrospective study to investigate the association between physical activity at the time of recruitment and weight gain from diagnosis among Korean breast cancer patients. They measured physical activity using IPAQ and reported that low physical activity was not significantly related to weight gain. Yaw et al. [36] conducted a retrospective study to investigate the association between changes in physical activity and weight change in Malaysia. They measured weight change using a weight scale and physical activity using a questionnaire from a year preceding breast cancer diagnosis to the study entry date. The mean time from diagnosis was 4.9 ± 3.5 years, and change in physical activity was not significantly related to weight gain. These previous studies were heterogeneous in study design and measurement methods. Therefore, further studies are required that utilize the same study design and measurement method.

Rock et al. also conducted a cross-sectional study among 1166 breast cancer patients, and they reported that higher dietary intake is independently associated with increased risk of weight gain [37]. Chen et al. conducted a prospective cohort study among breast cancer patients in China [12]. They measured dietary intake at baseline using a questionnaire and measured changes in body mass from a year pre-diagnosis to 18 months post-diagnosis. They reported that patients with a higher total dietary intake at baseline gained more body mass than those with a lower intake, but they did not adjust for meat and fish intake in their logistic regression analysis and did not report whether meat and fish intake was independent of weight gain. Lei et al. [38] conducted a prospective study to examine dietary changes among Chinese breast cancer patients. They measured dietary intake at baseline and at 36 months post-diagnosis using a questionnaire. The participants had significantly increased their vegetable and fruit consumption at 36 months post-diagnosis, from 4.5 to 5.6 servings per day. Conversely, the consumption of red meat, processed meat, and sugary drinks had decreased significantly. This indicates that dietary intake among breast cancer patients might become healthier after the diagnosis. However, caution should be taken when interpreting the results of the present and previous studies because each time point of the data collection was different.

Previous studies have described the mechanisms of diet-derived weight change. Naude et al. suggested that the total dietary intake is an important factor to explain the mechanism of weight change, but each food intake might also be related to weight change [39]. Vergnaud et al. stated that meat intake was positively associated with weight gain because of its higher energy density and fat content [40]. Thorsdottir et al. found that fish intake decreased more body weight than those who did not eat fish [41]. Although the mechanism between fish intake and weight change is unclear, a possible candidate is taurine, which is abundant in fish protein. A previous study reported that a taurine intake of 3 g per day for 7 weeks was associated with weight loss [42], but the number of studies reporting this is small, and further studies are needed. Mozaffarian et al. stated that vegetable consumption was inversely associated with weight change [43]. This result suggests that high vegetable consumption may replace higher energy density foods. The present study investigated the association of total dietary intake and each food intake with weight gain, but no significant association was found. Potential sampling bias and timing of measurement may have affected the results of the present study.

The present study found a significantly negative association between age and weight gain. Koo et al. [44] investigated the difference in weight gain between breast cancer survivors with a younger age group (18–54 years) and those with an older age group (55–75 years). The younger age group had a higher odds ratio for weight gain (OR: 2.5; 95% CI: 1.6–3.9) than the older age group. This previous study was consistent with our study results. A similar pattern of weight gain has been observed in the general (non-cancer) population of women [45]. Thus, age might be a factor related to weight gain; however, further studies are required to show the difference in the magnitude of weight gain between breast cancer patients and healthy women.

The major strength of this study was the objective evaluation of physical activity using an accelerometer. Most Asian studies used questionnaires for physical activity, but these tend to overestimate the amount of physical activity and involve recall bias [46]. Objective measurement of physical activity using accelerometers can overcome these limitations. Second, we evaluated not only endocrine therapy but also physical activity and dietary intake at the same time, as both factors might be related to weight gain. Therefore, evaluating these factors simultaneously is critical to address this issue accurately.

The present study had some limitations. First, as mentioned above, we measured physical activity and dietary intake at recruitment only. Breast cancer patients might have changed their lifestyles after diagnosis. It was necessary to investigate their lifestyles at the time of breast cancer diagnosis and afterward to understand the lifestyle of breast cancer patients better. Second, the present study might have had a sampling bias. More than 39% of the patients reported that their household income exceeded JPY 10 million per year. According to the Ministry of Health, Labour and Welfare, the median Japanese household income is JPY 5.6 million per year [47]. Furthermore, 45.5% of the patients had a university degree, but the average university enrollment rate among Japanese women was 16.1% at the time of the study (1999) [48]. Third, the response rate of patients in this study was low at 56%, which might cause a selection bias. Fourth, this study did not instruct patients to refrain from consuming food and drinks before weighing at diagnosis and recruitment since this study could not interfere with the regular medical check-up at the hospital. Fifth, this study did not investigate the existence of comorbidity and some medical characteristics (e.g., eating disorders, medications, and pregnancy). They might affect the results on the dietary intake. Sixth, in the present study, all patients who received endocrine therapy continued taking endocrine therapy based on the medical record. However, Kuda et al. reported that the adherence rate based on pill packets was 85% compared with the medical record (98%) [49]. We did not investigate the actual adherence rate of endocrine therapy using pill packets, and some patients might stop taking endocrine therapy. This might lead to a bias in the association between endocrine therapy and weight change and physical activity. Further studies are needed to address this issue. Seventh, we were not able to recruit an equal number of patients who received and did not receive endocrine therapy. This is because the original distribution ratio of patients who received and did not receive endocrine therapy was 7:3, and we noticed a difficulty in recruiting the two groups equally after the study commenced. Therefore, we decided to recruit more patients to maintain statistical power. As a result, the study population might have been biased, but few studies in Asia investigated the physical activity and dietary intake using validated indicators. This study’s findings may contribute to future research on this issue, but caution is required for generalization.

## 5. Conclusions

In summary, the present study examined the association between taking endocrine therapy, physical activity, or dietary intake and weight gain among Japanese patients diagnosed with breast cancer. Although we hypothesize that endocrine therapy is associated with weight gain, we did not find an association between endocrine therapy and weight gain in the present study. The reason for the non-significant association in the present study might be due to the much lower increase in weight gain among breast cancer patients compared with those of Western studies, and limitations such as the timing of measurement and sample bias may have affected the results of this study. Therefore, for these potential sampling biases and racial differences, caution is required for the generalization of this study’s findings.

## Figures and Tables

**Figure 1 medsci-09-00050-f001:**
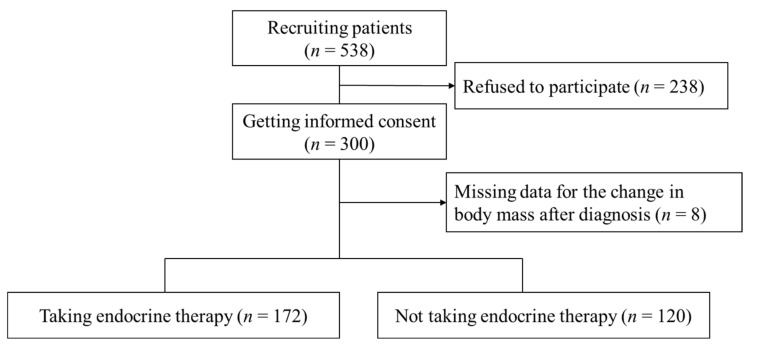
Flowchart of the study patients (Tokyo, Japan).

**Table 1 medsci-09-00050-t001:** Patient characteristics at the diagnosis of breast cancer.

Variable at Diagnosis	Overall *n* = 292	Taking Endocrine Therapy *n* = 172	Not Taking Endocrine Therapy *n* = 120	*p*-Value
Age at diagnosis (years)	46.6 ± 6.9	47.0 ± 6.3	46.2 ± 7.6	0.37
Body mass (kg)	53.7 ± 7.8	54.4 ± 8.5	52.7 ± 6.5	0.049
Height at surgery (cm)	158.6 ± 5.1	158.6 ± 5.0	158.6 ± 5.1	0.998
Body mass index (kg/m^2^)	21.3 ± 2.9	21.6 ± 3.1	20.9 ± 2.6	0.057
Underweight (< 18.5)	41 (14.0%)	21 (12.2%)	20 (16.7%)	0.20
Normal (18.5 to < 25)	219 (75.0%)	128 (74.4%)	91 (75.8%)	
Overweight (≥ 25)	32 (11.0%)	23 (13.4%)	9 (7.5%)	
Breast cancer stage (number)				
0	51 (17.5%)	2 (1.2%)	49 (40.8%)	< 0.01
I	99 (33.9%)	71 (41.3%)	28 (23.3%)	
II	115 (39.4%)	84 (48.8%)	31 (25.8%)	
III	27 (9.2%)	15 (8.7%)	12 (10.0%)	
Subtype (number)				
Luminal	198 (68.3%)	150 (88.2%)	48 (40.0%)	< 0.01
Triple positive	20 (6.9%)	18 (10.6%)	2 (1.7%)	
Her2	28 (9.7%)	0 (0%)	28 (23.3%)	
Triple negative	44 (15.1%)	2 (1.2%)	42 (35.0%)	
Chemotherapy				
Yes (number)	139 (47.6%)	83 (48.3%)	56 (46.7%)	0.79
No (number)	153 (52.4%)	89 (51.7%)	64 (53.3%)	
Radiation therapy				
Yes (number)	194 (66.4%)	120 (69.8%)	74 (61.7%)	0.15
No (number)	98 (33.6%)	52 (30.2%)	46 (38.3%)	
Type of endocrine therapy (number)				
Tamoxifen		99 (57.6%)		
Aromatase inhibitor		30 (17.4%)		
Combined		43 (25.0%)		
Postmenopausal (number)	73 (25.0%)	43 (26.2%)	30 (25.9%)	0.95

Note: age at diagnosis and body mass index are presented as mean and standard deviation; other data are presented as number and ratio. *p*-values were calculated using student’s *t*-test for continuous variables, and the chi-square test was used for categorical variables.

**Table 2 medsci-09-00050-t002:** Patient characteristics at recruitment.

Variable at Recruitment	Overall *n* = 292	Taking Endocrine Therapy *n* = 172	Not Taking Endocrine Therapy *n* = 120	*p*-Value
Time since diagnosis (year)	3.0 (3.0–5.0)	3.0 (3.0–5.0)	3.0 (2.0–5.0)	0.17
Body mass (kg)	55.0 ± 9.1	55.7 ± 9.5	54.0 ± 8.5	0.12
Body mass index, mean (standard deviation)	21.9 ± 3.5	22.1 ± 3.5	21.5 ± 3.5	0.13
Underweight (< 18.5)	39 (13.4%)	18 (10.5%)	21 (17.5%)	0.19
Normal (18.5 to < 25)	205 (70.2%)	123 (71.5%)	82 (68.3%)	
Overweight (≥ 25)	48 (16.4%)	31 (18.0%)	17 (14.2%)	
Weight change from diagnosis (kg)	1.3 ± 3.9	1.3 ± 4.0	1.4 ± 3.8	0.92
Weight gain more than 5% (number)	90 (30.8%)	57 (33.1%)	33 (27.5%)	0.31
MVPA using IPAQ (min/week) ^a^	300.0 (180.0–615.0)	307.5 (185.0–660.0)	300.0 (150.0–500.0)	0.26
Objectively measured MVPA (min/week) ^b^	462 ± 186	466 ± 181	456 ± 193	0.68
Objectively measured sedentary behavior (min/d) ^b^	503 ± 109	502 ± 110	505 ± 107	0.83
Dietary intake (kcal) ^c^	1643 ± 479	1645 ± 500	1641 ± 450	0.94
Fish and shellfish (kcal) ^c^	124.5 ± 88.2	118.6 ± 83.7	133.0 ± 93.9	0.18
Meat (kcal) ^c^	138.3 ± 80.9	139.1 ± 82.8	137.2 ± 78.4	0.84
Total vegetables (kcal) ^c^	72.7 ± 42.5	71.8 ± 40.0	74.4 ± 46.0	0.66
Smoking habit (number) ^c^				
Never	211 (75.6%)	122 (74.4%)	89 (77.4%)	0.73
Former	64 (22.9%)	39 (25.6%)	25 (22.6%)	
Currently	4 (1.4%)	3 (1.8%)	1 (0.9%)	
Education (number) ^c^				
High school	35 (12.5%)	24 (14.6%)	11 (9.6%)	0.20
Some college	117 (41.9%)	72 (43.9%)	45 (39.1%)	
≥ University	127 (45.5%)	68 (41.5%)	59 (51.3%)	
Postmenopausal (number) ^c^	197 (70.6%)	128 (78.0%)	69 (60.0%)	< 0.01
Household income (number) ^d^				
< 5 million yen	56 (20.3%)	34 (20.7%)	22 (19.6%)	0.89
≥ 5 million yen, < 10 million yen	11 (40.2%)	64 (39.0%)	47 (42.0%)	
≥ 10 million yen	109 (39.5%)	66 (40.2%)	43 (38.4%)	

Note: IPAQ, International Physical Activity Questionnaire; MVPA, moderate- to vigorous-intensity physical activity. Time since diagnosis and MVPA using IPAQ are presented as median and quartile. Body mass, body mass index, weight change from diagnosis, objectively measured MVPA and sedentary behavior, and dietary intake are presented as mean and standard deviation. All other data are presented as number and ratio. *p*-values were calculated using Mann-Whitney *U*-test, Student’s *t*-test, and chi-square test for non-parametric, parametric, and categorical variables. ^a^ Valid data were collected for 280 patients (164 received endocrine therapy and 116 did not receive endocrine therapy). ^b^ Valid data were collected for 276 patients (160 received endocrine therapy and 116 did not receive endocrine therapy). ^c^ Valid data were collected for 282 patients (166 received endocrine therapy and 116 did not receive endocrine therapy). ^d^ Valid data were collected for 279 patients (166 received endocrine therapy and 113 did not receive endocrine therapy).

**Table 3 medsci-09-00050-t003:** Odds ratio for 5% weight gain after diagnosis.

Variable	Crude Odds Ratio (95% CI)	Adjusted Odds Ratio (95% CI)
Age at diagnosis (year)	0.95 (0.92–0.989)	0.94 (0.89–0.99)
Time since diagnosis (year)	1.04 (0.92–1.19)	0.997 (0.85–1.17)
Body mass index at diagnosis (kg/m^2^)		
Underweight (< 18.5)	Ref	Ref
Normal (18.5 to < 25)	0.52 (0.26–1.04)	0.46 (0.20–1.06)
Overweight (≥ 25)	1.10 (0.43–2.80)	0.88 (0.26–2.96)
Breast cancer stage		
0	Ref	Ref
I	1.99 (0.91–4.36)	1.73 (0.59–5.06)
II	1.40 (0.64–3.07)	1.15 (0.33–4.07)
III	2.91 (1.06–7.99)	2.03 (0.43–9.61)
Chemotherapy		
No	Ref	Ref
Yes	1.39 (0.85–2.29)	1.43 (0.61–3.35)
Endocrine therapy		
No	Ref	Ref
Yes	1.31 (0.78–2.18)	1.12 (0.55–2.27)
Radiation therapy		
No	Ref	Ref
Yes	1.27 (0.74–2.16)	1.23 (0.63–2.41)
Postmenopausal at diagnosis		
Yes	Ref	Ref
No	1.78 (0.95–3.33)	1.19 (0.46–3.12)
Smoking		
Never	Ref	Ref
Former & currently	1.34 (0.75–2.40)	1.06 (0.54–2.10)
Education		
High school	Ref	Ref
Some college	0.75 (0.35–1.63)	0.69 (0.28–1.67)
≥ College	0.51 (0.23–1.11)	0.43 (0.17–1.13)
Household income		
< 5 million yen	Ref	Ref
≥ 5 million yen, < 10 million yen	1.56 (0.76–3.21)	1.61 (0.73–3.67)
≥ 10 million yen	1.19 (0.57–2.48)	1.12 (0.47–2.68)
MVPA using IPAQ (min/week)		
Q1 (≤ 180)	Ref	Ref
Q2 (181–300)	1.27 (0.64–2.54)	−
Q3 (301–618)	0.50 (0.24–1.06)	−
Q4 (≥ 619)	0.80 (0.40–1.61)	−
Objectively measured MVPA (min/d)		
Q1 (≤ 327)	Ref	Ref
Q2 (328–439)	0.46 (0.22–0.97)	0.43 (0.18–1.03)
Q3 (440–565)	0.60 (0.29–1.23)	0.48 (0.21–1.11)
Q4 (≥ 566)	0.85 (0.42–1.70)	0.78 (0.34–1.81)
Dietary intake (kcal/d)		
Q1 (≤1334)	Ref	Ref
Q2 (1335–1602)	1.23 (0.60–2.52)	1.28 (0.52–3.14)
Q3 (1603–1874)	1.03 (0.50–2.13)	1.06 (0.40–2.77)
Q4 (≥ 1875)	1.05 (0.51–2.18)	1.43 (0.49–4.17)
Fish and shellfish intake (kcal/d)		
Q1 (≤62)	Ref	Ref
Q2 (63–106)	1.32 (0.65–2.66)	1.39 (0.60–3.23)
Q3 (107–156)	0.86 (0.42–1.76)	0.87 (0.37–2.06)
Q4 (≥ 157)	0.71 (0.34–1.50)	0.78 (0.30–2.04)
Meat intake (kcal/d)		
Q1 (≤ 82)	Ref	Ref
Q2 (83–131)	1.14 (0.56–2.31)	1.06 (0.45–2.46)
Q3 (132–173)	0.77 (0.37–1.61)	0.68 (0.27–1.73)
Q4 (≥ 174)	0.94 (0.46–1.92)	0.96 (0.38–2.45)
Total vegetable intake (kcal/d)		
Q1 (≤ 43)	Ref	Ref
Q2 (44–65)	0.78 (0.39–1.56)	0.67 (0.28–1.57)
Q3 (66–90)	0.63 (0.31–1.29)	0.73 (0.31–1.70)
Q4 (≥ 91)	0.60 (0.29–1.23)	0.59 (0.23–1.54)

Note: all data are shown as odds ratios and 95% confidence intervals. Adjusted for age at diagnosis, body mass index at diagnosis, breast cancer stage, chemotherapy, endocrine therapy, radiation therapy, time since diagnosis, postmenopausal status, smoking, education, household income, objectively measured MVPA, dietary intake, fish and shellfish intake, meat intake, and total vegetable intake. We did not consider age and time since diagnosis as a categorical variable but as a continuous variable.

## Data Availability

The data presented in this study are available on request from the corresponding author. The data are not publicly available due to the protecting privacy.

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
