# Peer review of "Association between Endocrine Therapy and Weight Gain after Breast Cancer Diagnosis among Japanese Patients: A Retrospective Cohort Study"

_medsci, 2021, doi:10.3390/medsci9030050_

Round 1
Reviewer 1 Report
Journal: Medical Sciences
Manuscript ID: medsci1256220
Title: Association between endocrine therapy and weight gain after breast cancer diagnosis among Japanese patients: A retrospective cohort study
Overview:
This is an interesting article evaluating the impact of endocrine therapy on weight gain in patients with breast cancer post diagnosis. This manuscript is well structured and presents rational methods and results. Although no difference was found between cohorts, this manuscript addresses a common question in the clinical setting.
Major comments:
- This article should be evaluated for English grammar. Suggest having a native English speaker review the article
- The design of this trial suggests that it is a prospective, observational trial, not a retrospective cohort study. Please address this inconsistency in the title and throughout the manuscript. It may be useful to include a study design diagram in the supplementary materials.
Minor comments:
- With regards to line 144, were patients deemed not to be wearing the accelerometer device excluded from analyses/trial? Please clarify this point in the methods. From the Intent-to-treat design, please indicate how many people were excluded from this group.
- On line 172, the authors note a difficulty in recruiting to the “two groups equally”. Please elaborate on this in the discussion.
- In Table 3, I am unclear as to what direction is favoured for the odds ratio for age at diagnosis (?younger, ?older, ?age range) and time since diagnosis as there is no reference category. Please clarify this.
- Suggest revision of the discussion. Numerous studies are cited to address a single point (ie: 4 studies are discussed in sentence form to highlight endocrine therapy and weight gain, when one summary sentence with 4 references could be used).
Author Response
Thank you for reviewing our manuscript and giving relevant comments. We have revised our manuscript based on the reviewer's comments, and the responses are included in the attached file. The revised sentences are shown in red font in the revised manuscript.

Reviewer 2 Report
This study examined the association between weight gain and endocrine therapy in survivors of breast cancer. The hypothesis that endocrine therapy is associated with weight gain was not supported.
Key strengths of this report include the focus on an Asian population, which has unique differences from Caucasian populations; use of an objective measure of physical activity (tri-axial accelerometer), and a well-powered and transparently reported study methods (e.g., describing the need to increase sample size due to endocrine therapy use practice patterns).
There are several opportunities to strengthen this report.
The subjects enrolled in this study were 3 years from their diagnosis. How were women who started endocrine therapy after diagnosis, but stopped their endocrine therapy due to side effects before enrolling in the study handled? It is plausible that if a causal effect of endocrine therapy on weight gain does exist and occurs quickly within 12-24 months, by 3 years, some women, particularly those with significant weight gain, might have stopped endocrine therapy. This would bias your results to the null and obscure a difference between groups when a true treatment effect exists. This should be described more fully in the paper.
In the text, it might be preferred to report the between group differences, rather than the means and standard deviations for each group. This will allow readers to understand the magnitude of difference and complement what is presented in the tables.
What cut points were used to define physical activity intensity? This should be cited. The subjects in this study were very physically active, with 66 minutes per day of objectively measured physical activity. Again, this may indicate that some subjects who experienced side effects, such as arthralgias from estrogen deprivation that would limit physical activity volume may have already stopped therapy before enrollment. It is unclear how these subjects were handled in the analysis.
A semantic but important point: subjects with a BMI>30 should be referred to as “having obesity” and not an “obese subject.” We would never call someone with cancer a “cancerous patient”. The same should apply to persons living with the chronic disease of obesity.
Author Response

(The authors gave the same response as above.)

Round 2
Reviewer 1 Report
Ok to proceed to publication. The reviewers have addressed my comments
Reviewer 2 Report
The authors have been responsive to my initial comments.
This manuscript is a resubmission of an earlier submission. The following is a list of the peer review reports and author responses from that submission.
Round 1
Reviewer 1 Report
Association between endocrine therapy and weight gain after breast cancer diagnosis among Japanese patients: A retrospective observational study
This is a retrospective cohort study to assess the effect of the use of endocrine therapy on weight gain.
Comments
I think this study has several drawbacks to be published.
- The correct design is a retrospective cohort study.
- They had a response rate of 56% which is very low, and the authors did not show if there were differences between those who accepted to participate and those who did not.
- They mentioned that the purpose of this study was to investigate the association between endocrine therapy and weight gain. In fact, they could have assessed the effect of endocrine therapy on gain weight.
- They did not explain why patients had to ask permission to their doctors for participating in the study.
- One of the inclusion criteria says “3) those not diagnosed with metastatic cancer.” However, I table 1 they included patients with clinical stage II , III and IV. It is important to mention that patients on clinical stage IIB correspond to invasive breast cancer.
- There is specific indication for using antiestrogens. However, the authors did not include the results from immunohistochemistry to assess if the percentage of patients with positive estrogen receptors differed by groups (those who used and not used chemotherapy).
- Instead of using a Cox regression for a longitudinal study, they used a logistic regression.
- It is not clear the methodology used for modeling by using the logistic regression. Table 3 shows adjusted odds ratios and none of the variables included in the model were statistically significant. It looks that they just added all variables tas if the authors just added all variables to the model without assessing if they were real confounders.
- They even mentioned in the abstract that there should be caution for generalization because of sampling bias. If results are biased, they should not be published.
Reviewer 2 Report
General comments
This study is providing useful insight into factors that may affect weight gain in individuals who are Asian and also have a history of breast cancer. This information is appreciated, as much of the literature focuses on European and American cancer survivors. Since physical activity habits and dietary intake differ among cultures, it is important to recognize that what is relevant for one population of cancer survivors may not be relevant for another. Hopefully there will be more future studies examining these and similar issues.
Having said this, I found this manuscript to be lacking in several crucial elements. I have described these in my comments below for each section of the manuscript. For example, the procedures used to standardize the measurement of body mass need to be explained in more detail. Other aspects of the participants’ medical and social history need to be reported (e.g. pregnancy/childbearing history, presence of comorbid conditions, presence of eating disorders, presence of conditions affecting nutritional intake, other medication use). More detailed information related to dietary intake, rather than just daily caloric intake, need to be reported. These factors should then be examined within the statistical analyses to determine if they were associated with body mass change. If these factors haven’t been examined, then this needs to be explained in the Limitations section at the end of the Discussion, along with a rationale for why they were not examined.
Abstract
- Please use the term “body mass” rather than “body weight.” Please make this change throughout the manuscript.
- “Breast cancer is this most common cancer” is too vague. For example, it is the most common cancer overall in Japan? The most common cancer in women in Japan? Since your readership will include individuals from all over the world, it is helpful to be more specific when talking about a cancer prevalence.
- Page 1, lines 18-19: The Purpose statement needs to include the study population (e.g. women with a history of breast cancer).
- Page 1, lines 21-22: How long after the initial body mass was the current body mass measured? This can be expressed as mean ± standard deviation.
- Page 1, lines 22-23: Please state the instrument(s) used to collect physical activity and dietary intake information.
- Page 1, line 25: Is this the overall change in body mass for the entire study sample? It would be helpful to state the change in body mass for the group that received endocrine therapy separately from the group that did not receive endocrine therapy. Then, give the p-value that the t-test gave you to indicate that the difference was not statistically significant.
- Page 1, lines 27-28: Please include r-values and p-values to indicate that these associations were not statistically significant.
- Page 1, line 28: Which confounding factors did you adjust for? Please mention them here.
Introduction
- The Introduction section is concise and to the point, which is good.
- Page 2, line 43: I recommend adding in some text that describes why endocrine therapy may be associated with weight gain. This will help the reader to have a clearer insight into potential underlying mechanisms.
- Page 2, line 50-51: Is the association between meat and fish consumption and weight gain independent of caloric intake? Adding some text about caloric intake and weight gain in breast cancer survivors would be helpful to the reader.
Methods
- Page 2, line 82: Please give the location of St. Luke’s International Hospital.
- Page 2, lines 81-87: Please describe if the eligible breast cancer survivors were both females and males, or only females.
- Page 2, lines 81-87: How did you determine if eligible participants were without severe cognitive impairment? Did you determine this from their medical charts? Did you determine this via a questionnaire? Please describe this.
- Page 2, lines 81-87: Were all forms of major cancer treatments (e.g. survey, chemotherapy, radiation therapy, other targeted therapies) also eligible for inclusion? Please briefly describe this.
- Page 2, lines 81-87: Were all forms of endocrine therapy acceptable? Please briefly describe this.
- Page 2, lines 81-87: Were patients excluded for any non-cancer reason, such as presence of another co-morbid condition , medication history, pregnancy, etc.? Please describe this, as various comorbid diseases and conditions can affect dietary intake, the ability to absorb nutrients, and body mass changes.
- Page 3, lines 96-99: Please describe what standardization procedures you used to measure participants’ body mass. For example, what time of day was body mass measured? Were participants instructed to refrain from consumption of any food, drink, or other substances prior to having their body mass measured? Were they instructed to have voided their bladder and bowels prior to having their body mass measured? Since body mass is your primary outcome variable, it is important to state the procedures that you used to ensure that this variable was measured in the most accurate and precise way possible.
- Page 3, lines 110-122: Did you collect any retrospective information about participant physical activity habits at the time of their diagnosis, or during the time that elapsed between diagnosis and the current study? Please describe this.
- Page 3, lines 124-132: Similarly, did you collect any retrospective information about participant dietary habits at the time of their diagnosis, or during the time that elapsed between diagnosis and the current study? Please describe this.
- Page 3, lines 124-132: Similarly, did you collect information regarding the presence of eating disorders (e.g. anorexia nervosa, bulimia, etc.)? Please describe this.
- Page 4, lines 149-163: Please describe if you performed tests to determine if your data was normally distributed.
Results
- I believe it would also be helpful to present overall and group difference related to number pregnancies and births, pre-existing conditions (e.g. HTN, other cardiovascular diseases, diabetes mellitus, pulmonary disease, among others), and other medications being taken (e.g. anti-hypertensives, medications to control insulin, medications to treat depression/anxiety, and the like).
- Related to my point above, I might suggest that you run another regression analysis to examine if pregnancy/childbirth history, presence of co-morbid conditions, and other medication use are associated with body mass changes.
- Did you perform any regression analyses to examine other relationships between dietary intake and change in body mass, other than caloric intake? For example, in the Introduction, you mention associations between meat and fish consumption with changes in body mass. Therefore, I was expecting to see multiple aspects of dietary intake examined in this study. Please provide this information, as it may shed insight beyond daily caloric intake.
Discussion
- Please include a deeper discussion of potential physiological mechanisms that may underlie body mass changes and endocrine therapy. Even though the results of your study didn’t support your hypothesis that endocrine therapy was associated with body mass change, you still need to discuss the underlying science. This information would give the reader insight to the scientific rationale behind your study’s hypothesis.
- Similarly, please include a deeper discussion of potential mechanism that may underlie body mass changes and dietary intake. On Page 8, you once again discuss what previous literature has found regarding meat and fish consumption. You also discuss what previous literature has found regarding fruit and vegetable consumption. Please include what this and previous literature has found regarding rationale behind why different dietary intake patterns may be associated with body mass changes. Again, this information would give the reader insight to the scientific rationale behind why you chose to examine this variable its potential association with body mass.
Conclusions
- Page 9, lines 294-298: The conclusion statement is rather vague. Please add some text to indicate why you did not find your hypothesized associations.